# OBJECT-AVEDIT: AN OBJECT-LEVEL AUDIO-VISUAL EDITING MODEL

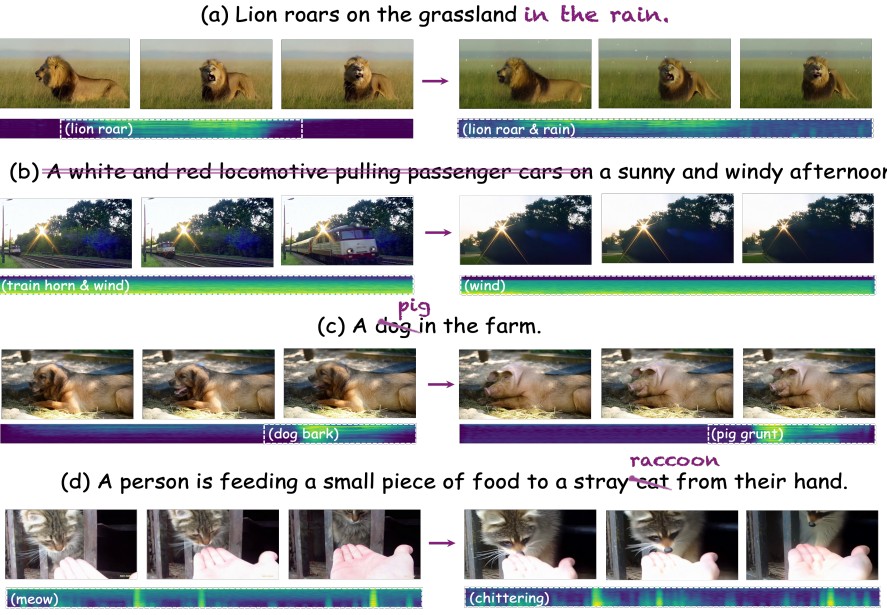

Figure 1: **Object**-**AVEdit** model provides object-level editing capability on audio-visual data. Users can implement the object-level editing operations like (a) **object addition**, (b) **object removal**, and (c, d) **object replacement** on audio-visual pairs with Object-AVEdit.

## ABSTRACT

There is a high demand for audio-visual editing in video post-production and the film making field. While numerous models have explored audio and video editing, they struggle with object-level audio-visual operations. Specifically, object-level audio-visual editing requires the ability to perform object addition, replacement, and removal across both audio and visual modalities, while preserving the structural information of the source instances during the editing process. In this paper, we present **Object-AVEdit**, achieving the object-level audio-visual editing based on the inversion-regeneration paradigm. To achieve the object-level controllability during editing, we develop a word-to-sounding-object well-aligned audio generation model, bridging the gap in object-controllability between audio and current video generation models. Meanwhile, to achieve the better structural information preservation and object-level editing effect, we propose an inversion-regeneration holistically-optimized editing algorithm, ensuring both information retention during the inversion and better regeneration effect. Extensive experiments demonstrate that our editing model achieved advanced results in both audio-video object-level editing tasks with fine audio-visual semantic alignment. In addition, our developed audio generation model also achieved advanced performance.

# 1 INTRODUCTION

Audio-visual data is an integral part of our daily lives, and natural language-guided object-level audio-visual editing shows immense potential due to its intuitiveness and efficiency in video post-production and filmmaking. Users often need the ability for precise object manipulation, for example, removing a dog and its accompanying bark from a scene, or replacing them with a pig and its sounds while leaving the background visuals and audio untouched. Many current models have explored editing on video or audio (Lin et al.; Wang et al., 2024; Lin et al.; Manor & Michaeli, 2024). But the object-level editing on audio-visual data has been overlooked. In this paper, we will focus on the object-level audio-visual editing, with operations mainly on three object-level fundamental editing tasks: **object addition**, **object replacement**, and **object removal**.

We adopted the inversion-regeneration editing paradigm (Hertz et al.) as our base, in which object-level editing relies on controlling the attention process of the target object using its text embeddings. This controllability is enabled by two key components: a word-level text encoder and an image-like encoding form (typically a Mel spectrogram for audio and video itself for video). The text encoder allows for the manipulation of attention scores based on the words describing the object, while the image-like encoding provides a spatial representation for the model to work with. Furthermore, high denoising quality is crucial for this controllability. The attention score for the objects must remain significant throughout the denoising process, enabling clear and effective object-level operations.

Meanwhile, the structural information preservation during the editing process and the regeneration quality are also important to the object-level audio-visual editing. Specifically, structural information preservation hinges on a non-information-loss inversion process, which is crucial for maintaining consistency between edited and original instances and keeping unedited zones unaltered. Similarly, regeneration quality dictates the overall object-level editing effect and the quality of the final output. A non-information-loss inversion and high-quality regeneration are essential for high-quality editing simultaneously.

Thus, we build our object-level audio-visual editing model prioritizing the following two aspects. **(a)** Unlike most video generation models, existing audio generation models (Liu et al., 2023; 2024b; Evans et al., 2025; Liu et al., 2025) lack the object-level controllability in the audio denoising attention processes, which is crucial for the object-level audio editing. To achieve the object-level audio-visual editing, we first developed a new audio generation model which has a clear correspondence between word-level text embeddings and sounding objects during the denoising attention process, enabling the object-level attention control required for object-level audio editing. **(b)** To preserve the structural information and achieve the better editing effect, we comprehensively considered the inversion-regeneration editing process, designed an inversion-regeneration editing algorithm optimizing the information retention during inversion and the regeneration quality simultaneously. By combining those, we propose the Object-AVEdit, achieving good audio-visual editing results as shown in Fig. 1. Our main contributions can be summarized as follows:

- We propose an **Object**-level **A**udio-**V**isual **Edit** model (Object-AVEdit), performing object-level high-quality addition, replacement and removal editing operations on both audio and video modalities.

- To achieve object-level audio editing, we developed a **new audio generation model**, which has an explicit correspondence between the word-level text embeddings and sounding objects in the audio during the denoising attention process, enabling the object-level attention control required for object-level audio editing.

- To ensure the structural information preservation and better editing effect, we designed a **inversion-regeneration holistically-optimized editing algorithm** to ensure both information retention during the inversion and the high-quality regeneration, which leads to the final high quality editing results.

Our model demonstrates advanced editing effects across both audio and visual modalities, with fine audio-visual semantic alignment in the edited audiovisual pairs. And our audio generation model also demonstrates advanced generation performance. Object-AVEdit can be widely applied in real-world video editing with sound, including filmmaking, short-form video production, and post-production.

## 2 RELATED WORK

### 2.1 AUDIO GENERATION AND EDITING

The audio generation and editing field has witnessed immense development. In audio generation field, AudioLDM (Liu et al., 2023), based on a CLAP language encoder (Elizalde et al.) and UNet architecture (Ronneberger et al., 2015), first achieved the audio generation task with fine effect. AudioLDM2 (Liu et al., 2024b) unified multiple conditional encoders, including CLAP, T5 (Kale & Rastogi) , and Phonemes encoder and supported multimodal inputs as the audio generation condition. Instead of encoding audio to Mel spectrograms, Stable Audio Open (Evans et al., 2025) directly encodes and denoises audio wave embedding, achieving good generation results in long time audio generation field. Recently, JavisDiT (Liu et al., 2025) trained an audio generation model with the T5 text encoder (Kale & Rastogi). At the same time, audio editing field has also made great progress. SDEdit (Meng et al., 2021) directly treats Mel spectrograms as images for editing, due to its lack of control over attention maps, it is difficult to guarantee the similarity between the edited and original audio. ZEUS (Manor & Michaeli, 2024), based on AudioLDM2 and DDPM Inversion (Huberman-Spiegelglas et al., 2024), can achieve sound replacement or unsupervised editing operation. However, due to the non-word-level text encoder of existing audio generation models, these methods still struggle to perform precise object-level editing. And experiments show the relatively poor denoising performance of JavisDiT audio generation model, making it challenging to be adapted to high quality audio editing.

### 2.2 VIDEO GENERATION AND EDITING

Research on video generation and editing models (Zheng et al., 2024; Team, 2024; Yang et al., 2024) has also achieved significant progress. Among the video generation models, Mochi-1 (Team, 2024) has a strong adaptability to real video domain (instead of animation or other unreal video domain). In video editing models, Stable V2V (Liu et al., 2024a) combines classical image processing techniques like object segmentation and depth estimation for frame-level video editing, while Video-P2P (Liu et al., 2024c) utilizes image generation models and the P2P method Hertz et al. (2022) for a similar purpose. RAVE (Kara et al., 2024) concatenates multiple video frames into a single image before applying image editing techniques. MotionDirector (Zhao et al., 2024) decouples the motion and appearance information from a video, using LoRA (Hu et al., 2022) to fit them separately to generate similar videos and RF-Edit (Wang et al.) innovatively combines the P2P method directly with advanced video generation models, enabling natural language to guide the editing process. However, these methods have low effect in the object-level editing operations due to crude model design. They either treat video as a series of disconnected images, making it difficult to maintain video continuity, or they lack high-quality editing process or fine-grained control. These factors limit the performance of existing video editing models.

## 3 METHOD

We will first present the fundamental preliminaries in Section 3.1. And we will elaborate the structure and training process of our audio generation model in Section 3.2, and the editing process in Section 3.3, with the attention control procedure within the editing process in Section 3.4.

### 3.1 PRELIMINARIES

In this section, we will introduce the notation used and the flow-matching based diffusion.

**Notations.** A video variable generally comprises the channels, frames, width, and height dimensions, which we use $x_v \in \mathbb{R}^{C \times F \times W \times H}$ to represent. In our model, we first transform the audio variable into Mel spectrograms, which have dimensions of channels, width, and height, and the channels dimension is always equal to 1. We use $x_a \in \mathbb{R}^{C \times W \times H}$ to represent audio variable. In most cases, the processing procedure is the same for both. Thus we use $x$ to represent them both. Consistent with current general generation model paradigm (Liu et al., 2024c; Manor & Michaeli, 2024; Wang et al.; Mokady et al.; Hertz et al.; Tumanyan et al., 2023), the editing process of our model is carried out in the latent space in our model. Variational Autoencoder (VAE), consisting of an encoder and a

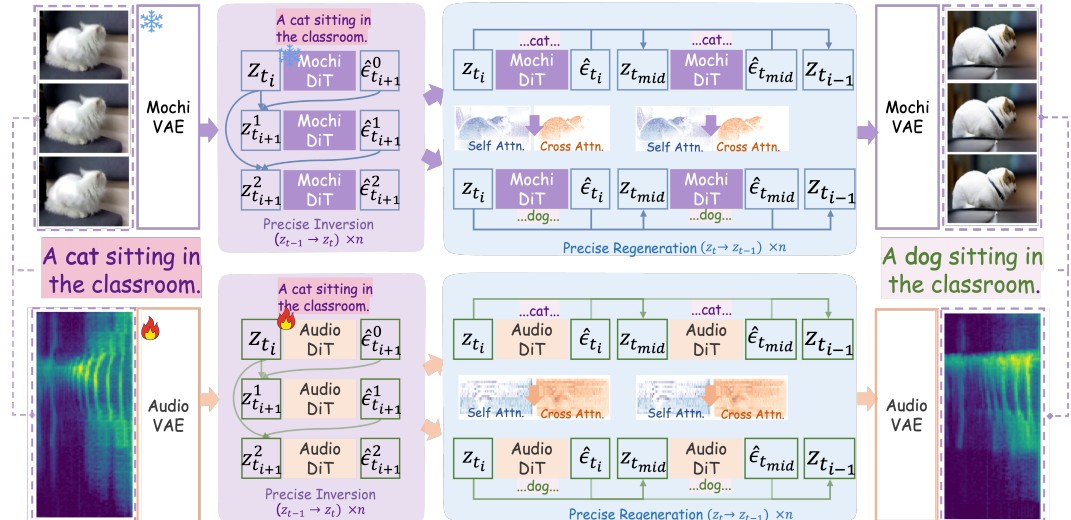

Figure 2: Editing pipeline of Object-AVEdit. The Object-AVEdit edits audio-visual data by first turning the original video and audio into noise. Then, the target prompt will be used to regenerate the semantically aligned edited video and audio, while preserving the original structure. In the regeneration process, our developed audio generation model is used to ensure the accessibility of the object-level attention maps. And inversion-regeneration holistically-optimized editing algorithm is applied to ensure both the structural information preservation during inversion and high regeneration quality.

decoder, will be used to transform the audio and video variable $x$ to latent space variable $z$, which will be inverted and generate the edited new variable $z^*$, and $z^*$ will be transformed back to real space to obtain the edited $x^*$. This process can be expressed as

$$z = \text{VAEencoder}(x), \tag{1}$$
$$x^* = \text{VAEdecoder}(z^*). \tag{2}$$

**Flow Matching.** In generation process, we use Flow Matching (Lipman et al., 2022) scheduler to estimate less noised variable $z_{t_{i-1}}$ from a noised variable $z_{t_i}$, with $z_1 \sim \mathcal{N}(0,1)$ and $z_0$ is the latent variable without noise. This process can be represented as

$$z_{t_{i-1}} = z_{t_i} + \int_{t_i}^{t_{i-1}} \hat{\epsilon}_\theta(z_t, t, c) dt, \tag{3}$$

where $c$ represents the condition for sampling and $\hat{\epsilon}_\theta(z_t, t, c)$ is learned to predict the velocity vector in the training process

$$\hat{\epsilon}_\theta = \arg\min_\theta \mathbb{E}_{z_0, z_1, t} \| \hat{\epsilon}_\theta(z_t, t, c) - (z_1 - z_0) \|^2. \tag{4}$$

Usually we approximate the second term of Eq. 3 using a first-order Taylor expansion, and the denoising process can be shown as

$$z_{t_{i-1}} = z_{t_i} + (t_{i-1} - t_i)\hat{\epsilon}_\theta(z_{t_i}, t_i, c). \tag{5}$$

**Flow Matching Inversion.** In the inversion-regeneration edit paradigm, we need to invert the unedited data $z_0$ to noise $z_1$. We can directly solve Eq. 5 to obtain the inversion process as

$$z_{t_i} = z_{t_{i-1}} + (t_i - t_{i-1})\hat{\epsilon}_\theta(z_{t_i}, t_i, c). \tag{6}$$

Given that the true value of $z_{t_i}$ is not available during inversion, it is a common practice (Xu et al., 2024; Manor & Michaeli, 2023; Song et al., 2020) to approximate it by $z_{t_{i-1}}$, which can be expressed as

$$z_{t_i} = z_{t_{i-1}} + (t_i - t_{i-1})\hat{\epsilon}_\theta(z_{t_{i-1}}, t_i, c). \tag{7}$$

After we got $z_1$ corresponding to the $z_0$, we can regenerate the $z'_0$, which we hope to be consistent with the original $z_0$ and the target $z_0^*$, which we hope to be aligned with the editing instruction, with the attention map control process to maintain the structural consistency between the original and edited latent.

## 3.2 Audio Generation Model Structure design and Training Process

The successful controllability for achieving audio editing process requires the correspondence between word-level text embeddings and the sounding objects to be edited in the audio. However, as explained in Section 1, current audio generation models (Liu et al., 2023; 2024b; Evans et al., 2025) are not well compatible. To address this problem, we developed a new audio generation model with explicit correspondence between the word-level text embeddings and sounding objects.

**Structure of Our Audio Generation Model.** Our developed audio generation model consists of VAE module (Kingma et al., 2013), T5 text encoder (Kale & Rastogi), DiT module (Peebles & Xie, 2023), and vocoder (Kong et al.). And mel spectrograms are used as the audio encoding form and Flow Matching (Lipman et al., 2022)-based generation scheduler is adopted. The specific module designs are detailed in the Appendix A. The design enables us to access the attention maps specific to the object to be edited and the attention maps specific to the object not to be edited, thereby enabling object-level controllability throughout the editing process.

**VAE Training.** Referring to LDM (Rombach et al., 2022), we use an adversarial learning paradigm to train the audio generation model. The overall loss includes a reconstruction loss $\mathcal{L}_1$, a regularization loss Kullback-Leibler loss $\mathcal{L}_{\text{KL}}$, and an adversarial loss $\mathcal{L}_{\text{GAN}}$. The total loss is summarized as

$$\mathcal{L}_{\text{VAE}} = \lambda_1 \mathcal{L}_1 + \lambda_{\text{KL}} \mathcal{L}_{\text{KL}} + \lambda_{\text{GAN}} \mathcal{L}_{\text{GAN}}. \tag{8}$$

**DiT Training.** We minimize the classic Flow Matching Loss (Lipman et al., 2022) during training, which can be expressed as

$$\mathcal{L}_{\text{DiT}} = \mathbb{E}_{\boldsymbol{z_0}, \boldsymbol{z_1}, t} \left\| \hat{\epsilon}_\theta(\boldsymbol{z_t}, t, c) - (\boldsymbol{z_1} - \boldsymbol{z_0}) \right\|^2. \tag{9}$$

By developing our audio generation model, we gain access to object-level attention maps during the editing process, which facilitates object-level controllability during audio editing.

## 3.3 Inversion-Regeneration Holistically-Optimized Editing Algorithm

As explained in Section 1, we adopt the inversion-regeneration editing paradigm (Hertz et al.) as our base, which requires the audio and video generation models with object-level controllability. We select Mochi-1 (Team, 2024) as our video generation model. Based on our developed audio generation model and Mochi-1, we can directly deploy the editing process according to the inversion-regeneration editing paradigm. The editing paradigm comprises an inversion and a regeneration phase. The inversion process transforms the original data (audio or video) into noise and the regeneration process denoises this noise latent to produce both the original object and the desired edited one. To ensure the edited one remains structurally consistent with the original one, an attention control strategy is employed during regeneration. In the following part of this section, we will elaborate how we ensure the structural information preservation and better editing effect by holistically optimizing both the inversion and regeneration process, while we will introduce the attention control process in Section 3.4. In our work, we achieve the structural information preservation inversion and high-quality regeneration in the editing process referencing to (Xu et al., 2024; Esser et al., 2021).

**Structural Information Preservation Inversion.** We utilize repeated inversion to make the inversion result closer to the corresponding true noise, achieving more precise inversion. Concretely, we first initialize $\boldsymbol{z}_{t+1}^0 = \boldsymbol{z}_t$ and iteratively apply the following equation to get a series estimation of $\{\boldsymbol{z}_{t+1}^k\}_{k=1}^K$ as

$$\boldsymbol{z}_{t_{i+1}}^{k+1} = \boldsymbol{z}_{t_i}^k + (\sigma_{t_{i+1}} - \sigma_{t_i})\hat{\epsilon}_\theta(\boldsymbol{z}_{t_{i+1}}^k, t_{i+1}). \tag{10}$$

Subsequently, we use

$$\boldsymbol{z}_{t_{i+1}} = \frac{1}{K} \sum_{k=1}^K \boldsymbol{z}_{t_{i+1}}^k \tag{11}$$

as the final value for $\boldsymbol{z}_{t_{i+1}}$.

**High-quality Regeneration.** We use the velocity vector predicted at an intermediate time step during sampling (e.g., from $t_i$ to $t_{i-1}$, we use the velocity vector $\hat{\epsilon}(z_{\frac{t_i+t_{i-1}}{2}}, \frac{t_i+t_{i-1}}{2})$ to approximate the average velocity vector $\frac{1}{t_i - t_{i-1}} \int_{t_i}^{t_{i-1}} \hat{\epsilon}(z_t, t)dt$, instead of $\hat{\epsilon}(z_{t_i}, t_i)$ in the classic sampling process)

to achieve the more precise sampling process. To get the value of the variable at an intermediate moment $\frac{t_i + t_{i-1}}{2}$, our sampling formulas are

$$
\begin{aligned}
t_{mid} &= \frac{1}{2}(t_i + t_{i-1}), \\
z_{t_{mid}} &= z_{t_i} + (t_{mid} - t_i)\hat{\epsilon}(z_{t_i}, t_i, C), \\
z_{t_{i-1}} &= z_{t_i} + (t_{i-1} - t_i)\hat{\epsilon}(z_{t_{mid}}, t_{mid}, C).
\end{aligned}
\tag{12}
$$

We integrate the precise inversion and high-quality regeneration algorithms to form our final editing algorithm. The pseudo-code for the complete editing process is shown in Appendix B.

### 3.4 ATTENTION CONTROL IN THE EDITING PROCESS

After we got the noise $z_1$ of the original $z_0$ as shown in Section 3.1 and Section 3.3, we will regenerate $z'_0$ consistent with $z_0$ under the prompt $\mathcal{P}$ of original data, and regenerate the desired edited data $z^*_0$ under the target prompt $\mathcal{P}^*$. In the regeneration, we control the attention process in denoising $z^*_0$ by editing its attention maps using the maps of $z'_0$ referring to (Hertz et al.). Assume that the self-attention maps and cross-attention maps at timestep $t$ when denoising $z'_0$ and $z^*_0$ are $M^s_t, M^c_t, (M^s_t)^*, (M^c_t)^*$. The editing process of the attention maps can be summarized as

$$
\begin{aligned}
\overline{M^c_t} &:= \begin{cases} (M^c_t)^* & \text{if } t < \tau^c, \\ (M^c_t)_{A(j)} & \text{otherwise,} \end{cases} \\
\overline{M^s_t} &:= \begin{cases} (M^s_t)^* & \text{if } t < \tau^s, \\ M^s_t & \text{otherwise,} \end{cases}
\end{aligned}
\tag{13}
$$

where $\overline{M^s_t}$ and $\overline{M^c_t}$ are the edited self-attention map and cross-attention map of $z^*_0$. The subscript $A(j)$ of $M^c_t$ represents the $A(j)$-th token-variable sub-cross-attention map of $M^c_t$. Here, $A(j)$ represents the position of the same word in $\mathcal{P}$ with the $j$-th word of $\mathcal{P}^*$. $\tau^s$ and $\tau^c$ represent the preservation strength of the self-attention map and cross-attention map. The roles of $\tau^s$ and $\tau^c$ are easy to follow. During the regeneration process, $t$ goes from 1 to 0. We apply the attention control in the initial period but deactivate it in the later stage.

Table 1: Preservation Strength in Video and Audio Editing Process.

| | Video | | | Audio | | |
|---|---|---|---|---|---|---|
| Metric | Addition | Replacement | Removal | Addition | Replacement | Removal |
| $\tau_s$ | 0.42 | 0.42 | 1.00 | 0.75 | 0.75 | 1.00 |
| $\tau_c$ | 0.42 | 0.42 | 0.42 | 0.75 | 0.75 | 0.75 |

## 4 EXPERIMENTS

In this section, we separately investigated the effectiveness of Object-AVEdit in both video and audio editing. Additionally, we also explored the performance of the audio generation model we developed.

### 4.1 DATASETS AND HYPERPARAMETERS

**Dataset** We use different existing datasets for training and evaluating. And to evaluate the object-level audio-visual editing effect, we construct **audio-visual editing datasets with object-level addition, replacement and removal tasks**. The details about our used datasets are provided in the Appendix C.

**Hyperparameters** We set the inversion and sampling steps of Mochi-1 to 64 and our audio generation model to 100 denoiseing steps in the editing process. The preservation strength of the attention map in the regeneration process are set as shown in Table 1.

### 4.2 COMPARISON BASELINES

We compare our Object-AVEdit with the state-of-the-art single-modality editing models.

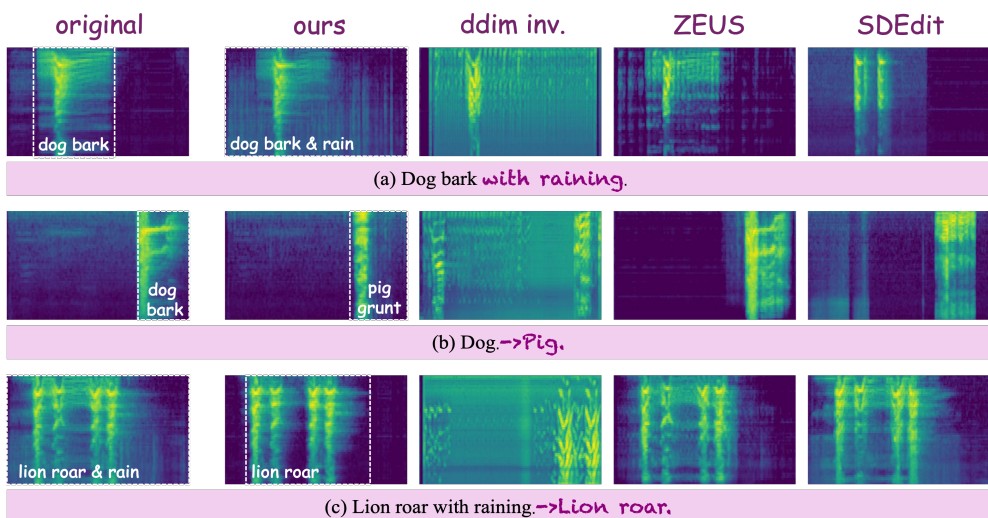

Figure 3: Performance of different audio editing methods on the addition, replacement and removal tasks. The prompts of the original audios and the desired edited audios are: (a) Dog bark. → Dog bark with raining. (b) Dog. → Pig. (c) Lion roar with raining. → Lion roar. From the Mel spectrograms, Object-AVEdit successfully edits the audio with preserving the structural information, which shows significant superiority.

**Audio editing evaluation.** We compare our model with ZEUS (Manor & Michaeli, 2024), DDIM Inv. (Song et al., 2020), and SDEdit (Meng et al., 2021). ZEUS is based on the DDPM inversion (Huberman-Spiegelglas et al., 2024), and SDEdit is based on the DDPM (Ho et al., 2020). The base model for these methods is AudioLDM2 (Liu et al., 2024b).

**Video editing evaluation.** We compare our model with RAVE (Kara et al., 2024) and RF-Edit (Wang et al., 2024). RAVE concatenates multiple frames into a single image and uses Stable Diffusion 2.1 (Rombach et al., 2022) as its base model. In the RF-Edit, the authors used Open-Sora (Zheng et al., 2024) as their base model. To ensure fairness in our experiments, we replaced its base model with Mochi-1.

**Audio-visual Semantic Alignment.** We compare our model with various combinations of basic video and audio editing models.

**Audio generation evaluation.** We compare our audio generation model with AudioLDM (Liu et al., 2023), AudioLDM2 (Liu et al., 2024b), and JavisDiT audio (Liu et al., 2025) to validate its generation performance.

### 4.3 METRICS

To evaluate the audio editing effect, we use CLAP (Elizalde et al.) (audio-text CLAP similarity of the edited audio) to evaluate the adherence of the edited audios to the editing commands and LPAPS (Iashin & Rahtu, 2021) (structural similarity between original and edited audios) to evaluate the structural consistency between the edited audios and the original ones. To evaluate the video editing effect, we use CLIP-T (mean frame-text CLIP (Radford et al.) similarity of the edited video) to evaluate the adherence to the editing commands, CLIP-F (mean inter-frame CLIP cosine similarity of the edited video) to evaluate the inter-frame consistency of the edited videos, and MUSIQ (Ke et al., 2021) (mean image visual quality) to evaluate the quality of the edited videos. To evaluate the audio-visual semantic alignment of different editing models, we use the Semantic Alignment Score (SAS). The SAS is defined as the mean cosine similarity between the ImageBind embeddings of the edited audio and video across all corresponding pairs in the results.

Note that CLAP and CLIP-T measure the adherence of the edited audios and videos to the editing commands. In addition and replacement tasks, the target objects is desired added or replaced to the

Table 2: Quantitative results of audio editing. Object-AVEdit achieves superior audio editing results with higher relevance to the target edits (CLAP) and better structural consistency (LPAPS) compared to existing methods across addition, replacement, and removal tasks.

| Method | Addition | | Replacement | | Removal | |
|---|---|---|---|---|---|---|
| | CLAP ($\uparrow$) | LPAPS ($\downarrow$) | CLAP ($\uparrow$) | LPAPS ($\downarrow$) | CLAP ($\downarrow$) | LPAPS ($\downarrow$) |
| before edit | -0.0737 | - | 0.0968 | - | 0.0243 | - |
| DDIM Inv. | **-0.0404** | 4.9149 | 0.0855 | 5.0989 | -0.0705 | 5.2884 |
| ZEUS | -0.0823 | 3.1665 | 0.1571 | 3.3416 | -0.0442 | 3.3922 |
| SDEdit | -0.1058 | 3.6643 | 0.1325 | 3.9673 | -0.0623 | 2.4444 |
| Ours | -0.0800 | **2.5700** | **0.2646** | **2.6330** | **-0.0866** | **2.4294** |

Table 3: Quantitative results of video editing. Object-AVEdit shows superior performance with higher inter-frame consistency (CLIP-F) and visual quality (MUSIQ) across addition, while also achieving strong relevance to the target edits (CLIP-T).

| Method | Addition | | | Replacement | | | Removal | | |
|---|---|---|---|---|---|---|---|---|---|
| | CLIP-T ($\uparrow$) | CLIP-F ($\uparrow$) | musiq ($\uparrow$) | CLIP-T ($\uparrow$) | CLIP-F ($\uparrow$) | musiq ($\uparrow$) | CLIP-T ($\downarrow$) | CLIP-F ($\uparrow$) | musiq ($\uparrow$) |
| before edit | 20.4091 | 0.9990 | 56.2400 | 23.7069 | 0.9961 | 53.8886 | 27.4256 | 0.9961 | 57.3884 |
| RAVE | 21.6931 | 0.9946 | 45.8112 | 25.4671 | 0.9917 | 51.0835 | 24.0076 | 0.9937 | 51.1930 |
| RF-Edit | **23.1053** | 0.9961 | 53.8744 | 24.5009 | 0.9946 | 46.4261 | 25.5004 | 0.9932 | 47.3331 |
| Ours | 21.9290 | **0.9971** | **62.1520** | **25.5853** | **0.9956** | 51.4918 | **23.2020** | **0.9971** | **53.1083** 6 |

edited audios or videos. Conversely, in removal tasks, the target object is to be removed. Consequently, CLAP and CLIP-T are positive indicators in addition and replacement tasks (higher is better), while they serve as negative indicators in removal tasks (lower is better, indicating successful removal of the target object).

### 4.4 AUDIO-VISUAL EDITING

In this section, we first evaluate the individual audio and video editing effects of Object-AVEdit, followed by an evaluation of its audio-visual semantic alignment.

**Audio Editing Effect** Results for different editing tasks achieved with the audio editing part along with a comparison to competing approaches are presented in Table 2. For fairness, We set total inversion and sampling steps to be 200 for all models and we kept the default settings of the adopted comparison models. For SDEdit, the noise level is set to $0.8$ (implying noise addition up to the timestep $t = 160$ out of 200). We make DDIM Inv. method start sampling from the 200-th timestep and ZEUS method start sampling from the 150-th timestep. Notably, different editing methods rely on various audio generation models, each with different optimal audio generation lengths and these audio models often perform well only on the audio lengths they are adapted to. Therefore, when using these models for audio editing, we first pad the audio to the length to which different models are adapted, and editing is then performed on this padded audio, and subsequently, the result is truncated back to the original length to ensure fairness in evaluating audio structural consistency. Overall, our model demonstrates superior results across all three editing tasks, significantly outperforming existing audio editing models. The editing effects of different models are visualized in Fig. 3.

**Video Editing Effect** Results for addition, replacement, and removal tasks achieved with our video editing part along with a comparison to competing methods are presented in Table 3. For fairness, we used a fixed sampling step of 64 for all methods. And for RF-Edit, we implemented their method on Mochi-1, ensuring consistency of the foundation model. As RAVE performs video editing based on image editing techniques, we followed the original setting and used Stable Diffusion 2.1 (Rombach et al., 2022) as its foundation model. Overall, our model demonstrates superior results across all three tasks, and our model significantly outperforming existing models in the removal and addition tasks. The editing effects of different models are visualized in Fig. 4.

**Audio-visual Semantic Alignment** We assessed the SAS of different editing models. As shown in Table 4, semantic alignment of Object-AVEdit is notably superior to the others, demonstrating the high-quality semantic alignment of its edited results.

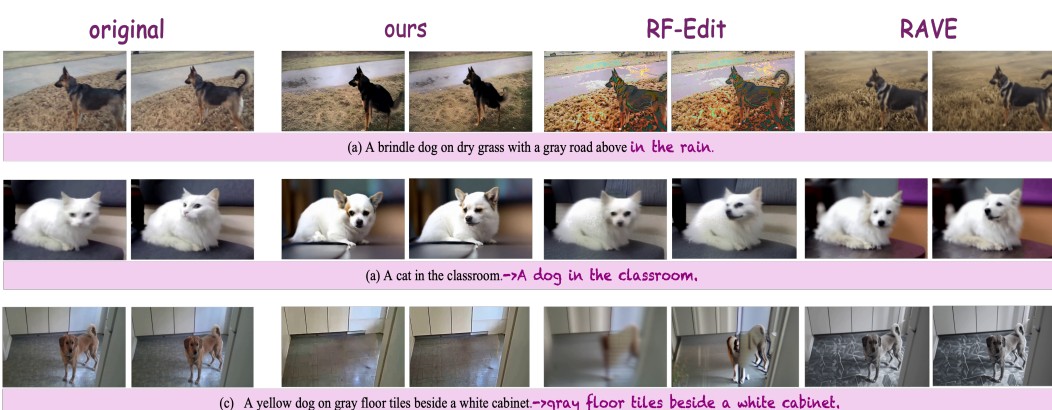

Figure 4: Performance of different video editing methods on the addition, replacement and removal tasks. The prompts of the original videos and the desired edited videos are: (a) A brindle dog standing on dry grass with a gray road above. → A brindle dog on dry grass with a gray road above in the rain. (b) A cat in the classroom. → A dog in the classroom. (c) A yellow dog on gray floor tiles beside a white cabinet → gray floor tiles beside a white cabinet. From the videos, Object-AVEdit achieves advanced effect in the object-level video editing tasks.

Table 4: Semantic Alignment Score of edited results between different models. Object-AVEdit shows notably superior to the others, demonstrating the advanced semantic alignment of edited results.

| Model | Ours | DDIM&RAVE | DDIM&RF-Edit | DDPM&RAVE | DDPM&RF-Edit | SDEdit&RAVE | SDEdit&RF-Edit |
|---|---|---|---|---|---|---|---|
| SAS (↑) | **0.3500** | 0.1340 | 0.1311 | 0.2496 | 0.2374 | 0.2110 | 0.2003 |

## 4.5 Audio Generation Models

Considering different audio generative models have different optimal audio generation lengths, we directly generate and evaluate audios at optimal generation lengths of each model in this experiment. As shown in Table 5, our developed audio generation model achieves advanced performance compared to current audio generation models, demonstrating superior semantic relevance to text prompts (highest CLAP score) and higher perceptual quality (FAD), while

Table 5: Quantitative results of audio generation. Our audio generation model demonstrates higher CLAP and FAD scores, while also competitive KL divergence.

| Method | CLAP(↑) | KL(↓) | FAD(↓) |
|---|---|---|---|
| GT | 0.3966 | - | - |
| AudioLDM | 0.2535 | 1.6365 | 0.3666 |
| AudioLDM2 | 0.3100 | 1.6371 | 0.1145 |
| JavisDiT audio | 0.2717 | **1.3827** | 0.1794 |
| Ours | **0.3473** | 1.4125 | **0.0945** |

also maintaining competitive feature distribution (KL) with ground truth audios. This guarantees a precise audio inversion and regeneration process, leading to effective editing results. In general, our audio generation model exhibits excellent compatibility with the inversion and regeneration editing paradigm and high quality of audio generation, providing a robust base for our audio editing process.

## 5 Conclusion

By training an advanced audio generation model and designing a precise editing algorithm holistically accounting for the inversion and regeneration editing processes, Object-AVEdit solves the following key problems in audio-visual editing: **a**. The inability of current audio generation models to deploy the inversion and regeneration editing paradigm for achieving high-quality object-level audio editing. **b**. The issue that previous editing methods only consider optimization of either the inversion or the regeneration stage. We proposed the Object-AVEdit in the paper, and it achieved advanced performance in the fields of object-level audio-visual data editing.

## 6 ETHICS STATEMENT

This research focuses solely on the technical advancement of object-level audio-visual editing, utilizing publicly available data and involving no human subjects or private information. While recognizing that powerful generative models inherently carry a risk of misuse for creating deceptive content (deepfakes), this work is strictly intended for academic exploration and responsible creative applications. We emphatically condemn any malicious use of the Object-AVEdit for generating misleading or harmful media, and we advocate for increased research into detection and provenance tracking to ensure the ethical deployment of such technologies.

## 7 REPRODUCIBILITY STATEMENT

We are committed to ensuring the full reproducibility of our work. Successfully replicating the results of the Object-AVEdit mainly hinges on two key components: the editing algorithm, which includes the inversion and regeneration processes, and the weights of our developed audio generation model. We detail the steps of our editing algorithm in Section 3.3 and provide its pseudocode in Appendix B. For direct access, the complete code of the editing process is available in the supplementary materials and will be open-sourced upon acceptance. The essential pre-trained weights for our audio generation model will also be made publicly available immediately following the acceptance of this paper.

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

## A    Detailed Information about Our Audio Generation Model

Our developed audio generation model consists of VAE module (Kingma et al., 2013), T5 text encoder (Kale & Rastogi), DiT module (Peebles & Xie, 2023), and vocoder (Kong et al.). Mel spectrograms are used as the audio encoding form and HifiGan (Kong et al.) is used as the vocoder, transforming the mel spectrograms back to the audio waves. Flow Matching (Lipman et al., 2022) scheduler is adopted. The depth, channels, number of trainable parameters of different model components and other detailed information about our audio generation model are shown as Table 6.

Table 6: Audio Generation Model Structure and Hyperparameters

| **Pre-processing** | | | |
|---|---|---|---|
| Sampling Rate | 16 kHz | Mel Channels | 64 channels |
| Mel Hop Length | 160 | Mel Frequency | 0-8k Hz |
| **DiT** | | | |
| Type | DiT (Peebles & Xie, 2023) | Depth | 96 layers |
| Hidden Size | 1024 | Parameter Count | 1.62 B |
| **VAE** | | | |
| Type | AutoencoderKL (Kingma et al., 2013) | Input/Output Channels | 1 channel |
| Latent Channels | 8 channels | Downsampling Factor | 4x4 |
| **Text Encoder** | | **Vocoder** | |
| T5 Text Encoder | T5 (large) (Kale & Rastogi) | Vocoder Type | HifiGan |
| Parameter Count | 716.8 M | Output Sampling Rate | 16 kHz |

## B    Pseudo-code of Precise Editing Algorithm

The pseudo-code of the precise editing process described in Section 3.3 is shown in Algorithm 1.

---

**Algorithm 1** Pseudocode for the complete editing process.

---

**Input:** Original latent $z_{t_0}$, inversion steps $N$, iteration steps $K$, Diffusion Model $\hat{\epsilon}$, source prompts $\mathcal{P}$, target prompts $\mathcal{P}^*$.

**Output:** Edited latent $e_{t_0}$.

1: **Phase 1: Inversion**
2: **for** $i \in \{1, 2, \ldots, N-1\}$ **do**
3:      $z_{t_i}^0 \leftarrow z_{t_{i-1}}$
4:      **for** $k = 1, \ldots, K$ **do**
5:          $z_{t_i}^k \leftarrow z_{t_{i-1}} + (t_i - t_{i-1}) \cdot \hat{\epsilon}(z_{t_i}^{k-1}, t_i, \mathcal{P})$
6:      **end for**
7:      $z_{t_i} \leftarrow \frac{1}{K} \sum_{k=1}^{K} z_{t_i}^k$
8: **end for**
9: $z_{t_N} \leftarrow z_{t_{N-1}} + (t_N - t_{N-1}) \cdot \hat{\epsilon}(z_{t_{N-1}}, t_{N-1}, \mathcal{P})$
10:   **return** $z_{t_N}$

11: **Phase 2: Generation**
12: $r_{t_N}, e_{t_N} \leftarrow$ output of line 10                  ▷ Initialize with the noisy latent from inversion
13: **for** $i$ in $\{N, N-1, ..., 2, 1\}$ **do**
14:      $t_{mid} = \frac{1}{2}(t_i + t_{i-1})$
15:      $r_{t_{mid}} = r_{t_i} + (t_{mid} - t_i)\hat{\epsilon}(r_{t_i}, t_i, \mathcal{P})$          ▷ Save Attention Map as $\text{Attn}_{t_{mid}}$
16:      $e_{t_{mid}} = e_{t_i} + (t_{mid} - t_i)\hat{\epsilon}(e_{t_i}, t_i, \mathcal{P}^*)$          ▷ Edit Attention Map using $\text{Attn}_{t_{mid}}$
17:      $r_{t_{i-1}} = r_{t_i} + (t_{i-1} - t_i)\hat{\epsilon}(r_{t_{mid}}, t_{mid}, \mathcal{P})$          ▷ Save Attention Map as $\text{Attn}_{t_{i-1}}$
18:      $e_{t_{i-1}} = e_{t_i} + (t_{i-1} - t_i)\hat{\epsilon}(e_{t_{mid}}, t_{mid}, \mathcal{P}^*)$          ▷ Edit Attention Map using $\text{Attn}_{t_{i-1}}$
19: **end for**
20: **return** $e_{t_0}$

---

## C  DATASET

We will detail the datasets used in training our audio generation model and evaluating the audio-visual editing effect and audio generation performance of different models.

**Datasets for training our audio generation model and evaluating the audio generation effect of different models** The datasets used for training our audio generation model include FSD50k (36k audios, 0.3-30s) (Fonseca et al., 2021), ClothoV2 (7k audios, 15-30s) (Drossos et al., 2020), AudioCaps (46k audios, 10s) (Kim et al., 2019), MACS (4k audios, 10s) (Martín-Morató & Mesaros, 2021), and VGGSound (200k audio-visual clips, 10s) (Chen et al., 2020). For FSD50k, AudioCaps and VGGSound, we directly utilize its provided text descriptions as their audio generation prompts. For ClothoV2 and MACS, which have multiple captions per audio, we paired each caption with its corresponding audio following the data process method in the training process of CLAP (Elizalde et al.). We utilize the AudioCaps evaluation set to assess the performance of audio generation models.

**Datasets for evaluating the effect of different audio and video editing methods** Given the limited editing tasks in existing audio and video editing evaluation datasetsLin et al.; Manor & Michaeli (2024), we introduce Object-AVEdit dataset, a dataset composed of audio-visual pairs with complex scenes and addition, replacement and removal editing tasks. All audio-visual pairs are with length of 3 seconds and are selected from VGGSound (Chen et al., 2020).

**Datasets for evaluating the effect of semantic alignment of edited audio and video pairs** For evaluating the semantic alignment of edited audio and video pairs, we created the Object-AVEdit-Alignment dataset. We curated this dataset by selecting samples from the Object-AVEdit dataset that required significant modifications in both the visual and audio modalities.

## D  AUDIO-VISUAL EDITING EFFECT OF OBJECT-AVEDIT

We demonstrate the effectiveness of Object-AVEdit on diverse examples. As shown in Figure 5, the model successfully performs audio-visual editing on various data.

## E  THE USE OF LARGE LANGUAGE MODELS (LLMS)

We used Large Language Models (LLMs) to aid and polish the writing of this manuscript, primarily for refining grammar, and clarity. The LLMs did not contribute to the research ideation, experimental design, data analysis, or the generation of the core scientific content of this paper (i.e., the proposed Object-AVEdit).

(a) An old, light blue sedan is driving through a rough, bumpy dirt field with a large willow tree *and a rusty tractor* in the background.

(b) A close-up of a vibrant African Grey parrot with its beak open, facing the viewer indoors *with a small colorful toy next to it*.

(c) ~~A small yellow bird perched on~~ a modern, diamond-shaped bird feeder against a gray background.

(d) ~~A brindle dog with~~ white cushions and a gray-blue carpet

(e) ~~A black steam train is running on~~ railway tracks under electrical lines leading towards a city skyline.

(f) Several brown ~~cows~~ *horses* are standing in a green alpine meadow under the tall, snowy mountains.

(g) A brindle ~~dog~~ *pig* with white cushions and a gray-blue carpet

Figure 5: Effectiveness of Object-AVEdit on diverse examples.

