# OpenReview forum: "Object-AVEdit: An Object-level Audio-Visual Editing Model"
_ICLR.cc/2026/Conference — Submitted to ICLR 2026_

### Official Review · Reviewer_wtuj · 2025-10-18

**Soundness:** 2
**Presentation:** 1
**Contribution:** 2
**Rating:** 4
**Confidence:** 5

**Summary:**

This paper presents an object-level audio-visual editing method in the inversion-regeneration paradigm. The proposed method is based on the inversion and [Prompt-to-Prompt](https://openreview.net/forum?id=_CDixzkzeyb) techniques. The authors trained a text-to-audio model that performs well, so that they can achieve great editing performance. The authors demonstrated that their proposed method works well in several aspects: audio editing performance, video editing performance, and audio-visual semantic alignment.

**Strengths:**

1. Although there is room for improvement in presentation, the paper itself is written well enough to make readers understand the proposed method and the experimental results.
2. The task this paper works on is interesting and important to the audio-visual or content creation community.
3. The authors quantitatively evaluated their method on their original dataset, which demonstrates the effectiveness of their proposed method. (The dataset would be useful, so I recommend making it publicly available in addition to the codebase.)

**Weaknesses:**

1. The proposed method is based on an existing image editing method ([Xu et al., 2025](https://openaccess.thecvf.com/content/CVPR2025/html/Xu_Unveil_Inversion_and_Invariance_in_Flow_Transformer_for_Versatile_Image_CVPR_2025_paper.html)). There is no big novelty in methodology.
2. Generated samples are not provided. Only spectrograms and video frames are provided in the form of images. Readers cannot grasp what edited videos are like in detail.
3. It would be good to evaluate audio-visual temporal synchronization, using the DeSync metric, whose implementation is provided at https://github.com/hkchengrex/av-benchmark .
4. In Section 4.5, the paper compares the trained text-to-audio model to only a few existing models. It lacks some SoTA-level text-to-audio models, such as [Make-An-Audio 2](https://github.com/bytedance/Make-An-Audio-2), [Tango 2](https://github.com/declare-lab/tango), [GenAU-Large](https://github.com/snap-research/GenAU), [ETTA-FT-AC-100k](https://github.com/NVIDIA/audio-intelligence/tree/main/ETTA), and [TangoFlux](https://github.com/declare-lab/TangoFlux).
5. It would be good to include ablation studies on the hyperparameters, which will help readers understand how sensitive the method is to hyperparameter values.
    - How performance changes according to the value of iteration step $K$. This ablation study will give an idea of how many iterations would be necessary.
    - How generated samples qualitatively vary according to the value of the inversion step $N$. Obviously, if $N$ is too small, a video will not be changed significantly. If $N$ is too large, a generated video will be largely different from the original one. However, readers will want to know the extent.

**Questions:**

I would appreciate the authors' responses to Weaknesses 3-5.

---

> ### Comment · Reviewer_wtuj · 2025-10-18
> **Minor comments**
>
> Let me suggest several modifications to make the manuscript more reader-friendly.
> - L.61: "Lin et al.; Wan et al.; Lin et al." -> "Lin et al." is repeated. The publication years should be mentioned.
> - L.65: "Hertz et al." -> The publication year should be mentioned.
> - L.113: "Elizalde et al." -> The publication year should be mentioned.
> - L.115: "Kale & Rastogi" -> The publication year should be mentioned.
> - L.136: "Hertz et al. (2022)" -> "(Hertz et al., 2022)" (\citep{} should be used here.)
> - L.161: "Mokady et al." -> The publication year should be mentioned.
> - L.206: "We can directly solve Eq. 5"
>   - This expression is confusing because we do not have access to the true $z_{t_i}$ and cannot adopt Eq.6, as explained in the paper. "We can transform Eq. 5" would be less confusing.
> - L.225: "Kong et al." -> The publication year should be mentioned.
> - LL.227-229: The details of the model design are not shown in the paper. Readers will not be convinced about this explanation.
> - L.258: "$z_{t+1}^0 = z_t$" -> "$z_{t_{i+1}}^0 = z_{t_i}$"
> - L.259: "$z_{t+1}^k$" -> "$z_{t_{i+1}}^k$"
> - Eq.10 (L.260): "$\sigma_{t_{i+1}}-\sigma_{t_i}$" -> "$t_{i+1}-t_i$"
> - L.370: "Radford et al." -> The publication year should be mentioned.

---

### Official Review · Reviewer_4ekj · 2025-10-22

**Soundness:** 2
**Presentation:** 2
**Contribution:** 1
**Rating:** 2
**Confidence:** 5

**Summary:**

This paper introduces Object-AVEdit, an object-level audio-visual editing framework aiming to perform targeted add / remove / replace operations on both the visual and acoustic modalities of a video. The method adopts an inversion–regeneration paradigm and proposes a “word-to-sounding-object” model to link textual cues with object-level audio generation. The paper targets video editing tasks that jointly modify sound and visuals while preserving background structures. While the topic is interesting and practically relevant, I find that the work suffers from several major weaknesses in terms of method design, experimental validation, and conceptual clarity.

**Strengths:**

- The attempt to unify visual and acoustic editing at the object level is timely and potentially impactful.
- The inversion–regeneration paradigm can, in principle, preserve structural consistency during edits.

**Weaknesses:**

1.	Unclear definition and verification of “object-level” control
- The paper claims to support object-level operations but does not define what constitutes an “object” (semantic entity, person, or background component?).
- The mechanism for object localization or segmentation is unclear. Without precise object boundaries, it is difficult to ensure that the model edits the target object only.
- Quantitative metrics assessing “object preservation vs. background distortion” are missing.

2.	Insufficient and unconvincing experiments
- The results are mainly shown as static frames. For a video + audio editing task, video demonstrations are essential to evaluate temporal consistency and audiovisual synchronization.
- The baselines are weak. The paper should compare with state-of-the-art visual editing models (e.g., object replacement / inpainting methods) and audio generation models.
- The user study design is unclear — sample size, blind evaluation, and fairness of questions should be reported.

3.	Limited methodological novelty
- The inversion–regeneration structure is well established in video editing. The proposed “word-to-sounding-object” module appears as an incremental engineering improvement rather than a fundamentally new idea.
- The coupling mechanism between audio and visual editing modules is under-explained: how is semantic alignment guaranteed, and how is temporal sync preserved?
- Overall, the approach feels more like a heuristic combination of existing components rather than a theoretically grounded framework.

**Questions:**

1.	Could you provide full video + audio examples for each editing type (add, remove, replace)?
2.	What baselines did you compare against? Are there object-level visual editing or audio editing baselines?
3.	How is the “word-to-sounding-object” model implemented and aligned with visual object representations?
4.	How do you quantitatively measure audiovisual synchronization after editing?
5.	What are the main failure cases (e.g., object removal artifacts, desynchronized sound)?

---

### Official Review · Reviewer_xiBt · 2025-10-30

**Soundness:** 2
**Presentation:** 2
**Contribution:** 2
**Rating:** 2
**Confidence:** 4

**Summary:**

The paper aims to perform object-level audio-visual editing. To achieve this goal, the authors train a diffusion-based audio generation model conditioned on word-level text embeddings, and apply the inversion-regeneration editing paradigm (Hertz et al.) to both audio and video. The method claims to enable object addition, replacement, and removal across modalities.

**Strengths:**

1. The paper targets an interesting setting: object-level joint editing of audio and video, which is relevant for audio-visual manipulation and post-production applications.

2. The writing is generally clear and the figures are well-presented, making the pipeline easy to follow.

**Weaknesses:**

1. Related work is not sufficiently discussed: The paper states that object-level audio-visual editing has been overlooked, but there are existing works on object-level audio editing, for example AUDIT (NeurIPS 2023) and other recent models that explicitly control sounding objects. These are not cited or discussed, and this weakens the novelty claim.

2. Lack of interaction between audio and video editing: The proposed method applies inversion-regeneration separately to audio and video. There seems to be no cross-modal linkage or constraint. This raises the question: How is this different from applying Hertz et al.'s paradigm independently to each modality? As presented, the only novelty appears to lie mainly in extending the paradigm to an audio diffusion model, rather than proposing a new joint editing technique.

3. The paper emphasizes  audio-visual object editing, but experiments appear to evaluate audio editing and video editing individually.  There is little analysis on whether the two modalities remain semantically aligned after editing (e.g., audio-visual consistency, coherence, or synchronization). Since this is essential to the claimed contribution, more justification is needed.

4. Baseline choices are outdated. The audio editing and audio generation baselines are relatively old. Several recent audio editing methods are not compared. Stronger baselines are needed to support the claim of state-of-the-art performance.

**Questions:**

1. What is the concrete methodological contribution beyond applying the inversion-regeneration paradigm to an audio diffusion model? In what sense is the approach new compared to Hertz et al.?

2. Since the method performs editing on audio and video independently, how is joint consistency ensured?

3. Can the authors provide demo examples of joint editing results? Audio-visual examples would make the claimed effects clearer.

**Details Of Ethics Concerns:**

No ethics concerns.

---

### Official Review · Reviewer_VVoi · 2025-11-03

**Soundness:** 2
**Presentation:** 2
**Contribution:** 2
**Rating:** 2
**Confidence:** 4

**Summary:**

This paper introduces Object-AVEdit, a framework for object-level audio-visual editing, designed to perform addition, replacement, and removal of objects and their corresponding sounds. The method operates on an inversion-regeneration paradigm. To enable object-level control, the authors first develop a new audio generation model based on a DiT architecture and a word-level T5 text encoder. They then propose an editing algorithm that incorporates techniques like repeated inversion and a higher-order sampler to enhance structural preservation and regeneration quality. The system is evaluated on a custom-built dataset, where it is shown to achieve promising results on various editing tasks compared to combinations of single-modality baselines.

**Strengths:**

The paper addresses a relevant and challenging problem: object-level audio-visual editing. The authors correctly identify a key limitation in existing audio models—the lack of fine-grained, word-level controllability—and make a commendable effort to build a new base model to address it. The overall structure of the paper is clear and easy to follow.

**Weaknesses:**

Despite the ambitious goal, the paper suffers from several major weaknesses in its methodology, evaluation, and positioning within the existing literature. These issues are significant enough to question the validity and contribution of the work.

1.  **Fundamentally Decoupled Architecture Contradicts "Audio-Visual" Claim:** The most significant flaw lies in the core architecture. Despite being titled an "Object-Level **Audio-Visual** Editing Model," the proposed pipeline (as shown in Figure 2 and described in Section 3) treats the audio and video modalities as two **completely independent streams**. The video is edited using Mochi-1, and the audio is edited using the authors' new model. There appears to be no cross-modal interaction, feature fusion, or joint-denoising mechanism between the two processes. This design is less an "audio-visual model" and more a "system that bundles two separate single-modality editors." This decoupled approach is a major technical limitation, as it cannot handle edits requiring tight temporal synchronization or complex audio-visual interplay. The claim of being an "audio-visual" model is therefore misleading.

2.  **Critically Incomplete Literature Review and Baseline Comparison:** The paper demonstrates a serious lack of awareness of the current state-of-the-art in the audio generation and editing fields. Numerous highly relevant and recent works are omitted:
    *   **Audio Editing:** Key works like **`AUDIT` (Wang et al., 2023)**, which focuses on instruction-following editing, and **`MEDIC` (Liu et al., 2024)**, which explores disentangled inversion for music editing, are not cited or compared against. These represent the SOTA in controllable audio editing. Other important baselines like **AudioEditor** are also missing.
    *   **Audio Generation:** The paper claims its new audio model achieves "advanced performance," yet it only compares against older models like AudioLDM/AudioLDM2. It fails to benchmark against more recent SOTA generators like *Tango2** **Stable Audio 2** , and **AudioLCM**, which would be necessary to truly validate its performance claims.
    This severe omission of related work undermines the paper's claimed novelty and the credibility of its experimental results.

3.  **Unverifiable Claims Due to Lack of Auditory Evidence (Demo Page):** For a paper centered on audio generation and editing, the absence of a supplementary demo page with audio examples is a fatal flaw. The quality of generated audio—including aspects like naturalness, artifacts, and successful preservation of background sound—cannot be assessed from static Mel spectrograms (Figure 3) or numerical tables. Without the ability to listen to the results and compare them directly with the baselines, the paper's core claims of "high-quality" editing and superiority over other methods are entirely unverifiable. This is a major departure from the standard practice for generative modeling research submitted to top-tier conferences.

4.  **Insufficient and Unconvincing Evaluation Metrics:** The evaluation of the audio editing task is inadequate and lacks rigor.
    *   **Lack of Subjective Evaluation:** The paper presents no formal subjective evaluation (e.g., a Mean Opinion Score - MOS study) to assess the perceptual quality of the edited audio. For a creative, perceptual task like audio editing, objective metrics like CLAP and LPAPS are known to be insufficient and often correlate poorly with human judgment.
    *   **Incomplete Objective Metrics:** The chosen metrics (CLAP, LPAPS) are indirect. The evaluation lacks a direct measure of **task success rate** or **editing accuracy**. For instance, an audio classification model or an AudioLLM could have been used to verify if a target object's sound was truly removed or added. Without such a metric, it is difficult to determine if the model is correctly performing the requested edit, or simply generating audio that is semantically close to the target prompt.

5.  **Ambiguous Problem Definition:** The paper fails to provide a clear, formal definition of "object-level" editing. The term is used throughout but is only explained via examples. What constitutes an "object"? How are overlapping sound objects handled? How are objects disentangled from background ambiance? The lack of a precise conceptual framework weakens the paper's scientific rigor.

**Questions:**

1.  **On the Necessity of the T5 Encoder:** Your newly developed audio model uses a T5 text encoder to achieve word-level control, which is key to your editing paradigm. Have you investigated the emergent object-level correspondence in models trained with sentence-level encoders like CLAP? Is it possible that even without explicit word-level supervision, some degree of object-level attention emerges that could be leveraged, perhaps with more sophisticated attention extraction techniques? What makes the T5-based explicit correspondence fundamentally superior to potentially implicit ones?

2.  **Robustness of Attention Control:** The editing process relies heavily on manipulating attention maps. How robust is this mechanism to semantic ambiguity in the prompts? For example, if the original prompt is "A dog is playing" and the target is "A cat is playing," the model must correctly identify that only the "dog" token's attention needs to be replaced by "cat," while the "playing" token's attention (which might relate to background sounds or motion context) should be preserved. How does your system handle such shared-concept words, and what happens if the model incorrectly assigns attention?

3.  **Scalability to Multi-Object Scenes:** The examples shown involve simple, single-object edits. How does the performance of your attention control mechanism scale as the number of sound-emitting objects in a scene increases? For instance, in a scene with "a dog barking, a bird chirping, and a car passing by," if the goal is to only replace the "dog" with a "cat," how do you ensure that the attention maps for the "bird" and "car" remain perfectly un-touched and do not "leak" into the edited region?

4.  **The Role of Inversion Quality:** Your paper emphasizes a "holistically-optimized" algorithm with high-quality inversion. Could you elaborate on the qualitative difference in editing outcomes when using a "lossy" inversion (like standard DDIM inversion) versus your high-fidelity one? Does poor inversion lead to specific types of artifacts (e.g., background corruption, color shifts, muffled audio), or does it simply result in a lower overall similarity to the original? Understanding this trade-off is crucial for practical applications where speed might be preferred over perfect reconstruction.

5.  **Exploring the Latent Space:** Since your model can invert real audio-visual data into a shared paradigm (the latent space of a diffusion model), have you explored other applications of this inverted latent space beyond attention-based editing? For example, could one perform arithmetic operations on the latents (e.g., `latent(dog) - latent(bark) + latent(meow)`) to achieve editing, and how would that compare to the attention-based approach? This could reveal deeper properties of the learned representations.

---

### Meta-Review · Area_Chair_wjM4 · 2026-01-06

**Summary:**

All reviewers believe that the paper lacks novelty, does not discuss related works sufficiently, and has insufficient and unconvincing experiments.

Specifically, reviewers raise concerns including:
1) the technical contribution of the paper, noting that the architecture appears to be a combination of existing tools rather than a novel framework;
2) a primary concern across all reviews is the incomplete literature review;
3) reviewers find the experimental results unconvincing;
4) despite the "audio-visual" branding, the reviewers argue that the paper does not sufficiently address or evaluate the alignment between modalities.

**Reviewer Concerns:**

The authors did not post rebuttal. All concerns remain unsolved as no rebuttal has been posted.

**Reviewer Scores:**

The reviewers will not change their scores as no rebuttal has been posted.

---

### Decision · Program_Chairs · 2026-01-26

Reject